# Co-Infection of the Epstein–Barr Virus and the Kaposi Sarcoma-Associated Herpesvirus

**DOI:** 10.3390/v14122709

**Published:** 2022-12-02

**Authors:** Michelle Böni, Lisa Rieble, Christian Münz

**Affiliations:** Viral Immunobiology, Institute of Experimental Immunology, University of Zürich, 8057 Zurich, Switzerland

**Keywords:** natural killer cells, cytotoxic lymphocytes, T cells, latent and lytic infection, B cell lymphomas, Kaposi sarcoma, humanized mice

## Abstract

The two human tumor viruses, Epstein–Barr virus (EBV) and Kaposi sarcoma-associated herpesvirus (KSHV), have been mostly studied in isolation. Recent studies suggest that co-infection with both viruses as observed in one of their associated malignancies, namely primary effusion lymphoma (PEL), might also be required for KSHV persistence. In this review, we discuss how EBV and KSHV might support each other for persistence and lymphomagenesis. Moreover, we summarize what is known about their innate and adaptive immune control which both seem to be required to ensure asymptomatic persistent co-infection with these two human tumor viruses. A better understanding of this immune control might allow us to prepare for vaccination against EBV and KSHV in the future.

## 1. Introduction to EBV and KSHV

The two human γ-herpesviruses, Epstein–Barr virus (EBV) and Kaposi sarcoma-associated herpesvirus (KSHV), are WHO class I carcinogens [1]. They are associated with lymphomas and carcinomas that fortunately only develop in a small percentage of persistently EBV and KSHV infected individuals [2,3,4,5]. EBV persists in more than 95% of the adult human population and KSHV is most frequent in Sub-Saharan Africa with a seroprevalence of more than 50% in many countries but below 10% in Northern Europe and Northern America [4,6]. Both viruses are thought to be primarily transmitted via saliva exchange, and infect B cells in submucosal secondary lymphoid tissues, such as tonsils [6,7]. EBV might cross the mucosal epithelium via transcytosis [8,9]. EBV establishes latent antigen expression after infection that drives B cells into proliferation and rescues them from cell death. This leads to B cell immortalization, as can be observed in vitro during the generation of lymphoblastoid cell lines (LCLs) by EBV infection of primary human B cells [10]. The latency III program that is found in LCLs consists of six EBV nuclear antigens (EBNA1, 2, 3A, 3B, 3C and -LP), two latent membrane proteins (LMP1 and 2), two small non-translated RNAs (EBER1 and 2) and more than 40 miRNAs. It can also be detected in naïve tonsillar B cells of healthy virus carriers [11]. In germinal center B cells, latent EBV protein expression is reduced to EBNA1, LMP1 and LMP2. This latency II program is thought to provide CD40 and B cell receptor (BCR)-like signaling to rescue infected B cells from the germinal center reaction. This differentiation allows EBV to gain access to the memory B cell pool in which all latent EBV protein expression is turned off (latency 0) or EBNA1 is transiently expressed to maintain the viral DNA in homeostatically proliferating memory B cells (latency I) [12,13]. From this reservoir of long-term persistence, EBV reactivates into lytic replication and infectious viral particle production, most likely due to BCR stimulation-induced plasma cell differentiation [14]. Accordingly, the viral transcription factor BZLF1 that initiates lytic EBV replication in B cells is induced by the plasma cell-associated transcription factors, BLIMP1 and XBP1 [15,16]. Basolateral infection of mucosal epithelial cells [8] might then allow for another round of lytic EBV replication as is pathologically observed during oral hairy leukoplakia [17] for efficient viral shedding into saliva and further transmission. Therefore, all latent EBV infection patterns that are found in B cell lymphomas, including latency I of Burkitt’s lymphoma, latency II of Hodgkin’s lymphoma and latency III that can be observed in some diffuse large B cell lymphomas (DLBCL), are already present in healthy EBV carriers. Immune suppression due to human immunodeficiency virus (HIV) co-infection or iatrogenic immune suppression after transplantation allows these premalignant states to develop into the respective lymphomas. For KSHV, the sites of latent and lytic infection are much less well defined. However, due to the emergence of Kaposi sarcoma (KS), primary effusion lymphoma (PEL) and multicentric Castleman’s disease (MCD) during immune suppression, KSHV persists and is presumably immune controlled in endothelia and B cells from which Kaposi sarcoma and the KSHV-associated lymphomas emerge [4,5]. How the three KSHV latent gene products, viral FADD-like interleukin-1-β-converting enzyme inhibitory protein (vFLIP), viral cyclin (vCyclin) and latent nuclear antigen (LANA), and its lytic gene products contribute to the non-pathogenic cellular reservoirs of persistent KSHV infection remains to be defined. However, recent studies suggest that at least some of these benefit from co-infection by EBV for KSHV persistence.

## 2. Persistence of KSHV in EBV Infected B Cells

KSHV infection has been associated with primary effusion lymphomas (PELs) since 1995, and KSHV detection has been an important part of the PEL diagnosis ever since [18,19]. Knockdown of LANA as well as vCyclin and vFLIP has led to growth inhibition and apoptosis in PEL cell lines, and leads to a reduction in KSHV genome levels [20]. Further, knockdown of the viral interferon regulatory factor 3 (vIRF3) has also been shown to reduce proliferation of PEL cells and increase apoptosis levels [21]. All this supports the association of PEL with KSHV infection.

In addition to KSHV, about 90% of PELs show persistence of EBV [22,23,24]. Co-infection is frequently detected in established PEL cell lines, with both viral genomes maintained and independently replicated and partitioned to the daughter cells [22,25,26]. In vitro studies showed that KSHV alone can infect but not transform peripheral B cells and therefore cannot persist long term [27,28,29]. In vivo dual-infection studies in mice with reconstituted human immune system components (humanized mice) have added evidence that co-infection with EBV increases the probability of KSHV persistence [30,31]. Co-infection with EBV activates B cells and supports long-term KSHV infection and cell proliferation through transformation depending on expression of at least one transforming EBV gene [25,27]. Persistence of KSHV is not dependent on EBV lytic gene expression, as KSHV can also persist in cells infected with an EBV BZLF1 knockout virus that lacks lytic gene expression in vitro and in vivo [30].

B cell transformation has been shown to be dependent on five viral latent antigens, namely EBNA2, EBNA-LP, EBNA3A, EBNA3C and LMP1. Proliferation of infected cells is initiated by EBNA2 through expression of cell cycle genes such as c-myc and cyclins D2 and E [32,33]. EBNA-LP is reported to enhance this EBNA2-induced gene activation [34,35]. EBNA3A and EBNA3C block the DNA damage response; however, animal experiments have shown that they are not necessary for EBV persistence [32,33,36]. LMP1 expression contributes to transformation and proliferation as well as cell survival by engaging NF-κB signaling pathways and mimicking CD40 signaling [37,38,39]. EBV LMP1 supports latency establishment through inhibition of lytic replication, and transcriptional control in PEL allows for sporadic expression of LMP1 and non-coding RNAs [40,41,42,43,44,45,46]. Expression of these gene products creates conditions permitting KSHV persistent infection and PEL emergence [27].

The link of EBV and PEL proliferation is further supported by the fact that EBV genome loss reduces both KSHV genome maintenance and proliferation [25,27,47]. The exact mechanism is unknown, but EBV co-infection seems to maintain KSHV genomes, as evident by the increased amount of KSHV genomes per cell observed in co-infected cells [25,27,30,48]. Later during co-infection, expression of both EBV and KSHV is restricted to a reduced latent EBV gene expression. It is mostly restricted to EBNA1 and non-translated RNAs (latency I), as KSHV LANA induces methylation and silencing of the major latent promoters Qp and Cp that regulates expression of EBV latency III genes [49].

While EBNA1 might only contribute little to B cell transformation, loss of its expression in EBV^+^ KSHV^+^ PEL cell lines reduced proliferation, indicating a role of EBNA1 in the promotion of KSHV persistence and B cell growth [40,47]. 

Aside from EBV genes, PEL cells depend on latent KSHV gene expression, mainly LANA, vFLIP and vIRF3 for survival, as they interact with tumor suppressors and inhibit apoptotic processes [20,21,50,51,52]. LANA mediates the persistence of the KSHV episome by interaction with KSHV terminal repeat sequences [53,54]. This persistence does not depend on further viral genes, and episomes are lost upon LANA knockdown [55]. LANA also mediates replication of the episomal DNA and tethers the virus DNA to host mitotic chromosomes, facilitating division of the KSHV genome to the daughter cells [56]. vFLIP can activate NF-κB, which is constitutively active in PEL [57,58,59]. It averts FAS-induced apoptosis through interaction with the death-inducing signaling complex (DISC) that prevents processing of procaspase 8 [60]. vIRF3 is required for survival of both EBV^+^ and EBV^−^ PEL as knockdown lead to an increase in apoptosis and reduced proliferation [21]. 

Apart from these molecular interactions of EBV and KSHV gene products for persistence of both viruses in B cells in vitro and in mouse models, epidemiological evidence in Sub-Saharan Africa has suggested that KSHV infection is nearly uniformly associated with EBV co-infection and that EBV seropositivity is among the strongest environmental risk factors for KSHV seropositivity [61,62]. Therefore, EBV gene expression contributes to the persistence of KSHV in B cells, promoting B cell transformation, proliferation and survival. This allows for KSHV persistence due to EBV co-infection in vitro, in mouse models and in a human African patient cohort.

## 3. Primary Effusion Lymphomagenesis Due to EBV and KSHV Co-Infection

As EBV increases KSHV persistence, KSHV genome copy numbers per cell and cell proliferation, it is highly likely that it also impacts primary effusion lymphomagenesis. Development of primary effusion lymphoma is still not completely understood, but in vitro and recent in vivo studies suggest a role of viral lytic gene expression in driving tumorigenesis [30,63].

EBV and KSHV dual-infected humanized mice present with increased lymphomagenesis and enhanced levels of early EBV lytic gene expression [30,64,65]. These enhanced levels of EBV lytic gene expression are also detected in co-infected PELs, supporting the role of lytic genes in tumorigenesis [16,30,66,67]. Infection with BZLF1-deficient EBV demonstrated a reduction in lymphoma formation, whereas infection with an EBV variant that increases lytic replication demonstrated increased lymphomagenesis compared to EBV wildtype infection in humanized mice [64,68,69]. It is likely that the increase in tumor formation is promoted by abortive lytic EBV expression, as full lytic EBV reactivation would rather decrease tumor formation by the destruction of infected cells during the production of new viral particles [6,70]. Expression of BZLF1 induces lytic gene expression as well as the expression of immune evasins and proteins protecting the cells from apoptosis [71]. BZLF1 itself has been shown to play a prominent role in tumor progression through its capability to induce VEGF and IL10 secretion (Figure 1), supporting vascularization and suppressing T cell responses [72,73,74,75,76]. Lack of the late lytic gene BALF5 increases establishment of lymphomas from transformed B cells in immunocompromised mice, confirming a role of early lytic genes [77]. In EBV^+^ B cells, tumor necrosis factor (TNF), CCL5 and IL10 expression is increased upon spontaneous lytic reactivation [78,79,80]. This links lytic EBV expression to conditioning of the tumor microenvironment, as TNF is involved in inflammation and immune regulation, CCL5 is important in the recruitment of myeloid suppressor cells and IL10 suppresses T cell responses [78,79,80]. Adding to this, EBV itself encodes for a viral homologue of IL10 (vIL10) [81]. KSHV encodes for a viral homologue of IL6 (vIL6) that in turn can upregulate production of human IL6 and IL10 [82]. vIL6 cooperates with c-myc and drives formation of plasmablastic neoplasms in immunocompromised mice, as well as it increased the number of tumors in a murine xenograft model and supported metastasis [83,84,85,86]. These cytokines increase the production of Vascular Endothelial Growth Factor (VEGF) and together, this promotes proliferation, cell survival, immunosuppression, neoangiogenesis and activation of oncogenic signaling pathways such as the NF-κB pathway [82,87,88,89,90]. 

Many studies demonstrate an important role for a multiplicity of KSHV genes in lymphomagenesis. ORF36, a viral protein kinase, leads to increased hyperproliferation of B cells as well as lymphoma development [91]. Transgenic expression of the transmembrane glycoprotein K1 promotes lymphoproliferations that show NF-κB activation [92,93]. K1 can also induce expression of VEGF and pro-inflammatory cytokines like IL6, IL8 and IL10 [93,94,95]. Viral G-protein coupled receptor (vGPCR) increases expression of pro-inflammatory cytokines and contributes to tumor formation that resembles Kaposi sarcoma when expression is induced in mice [96,97,98,99,100]. 

vIRF3 drives an oncogenic transcriptional program mediated by super-enhancers through cooperation with cellular IRF4 and BATF [21,43]. RTA, the replication and transcription activator of KSHV, can transactivate EBV latency promoters by complexing with RBP-Jκ [44]. This cooperation induces LMP1 expression in an EBV latency I background, contributing to cell growth that is EBV-driven [44]. It further interacts with the EBV lytic inducer BZLF1, inhibiting EBV lytic gene expression [44,45,101]. LMP1, in turn, contributes to tumor formation through inducing expression of the oncogenic protein UCH-L1 [102]. The latent KSHV gene LANA has also been shown to induce UCH-L1, and co-infection has shown that LANA and LMP1 synergize to activate UCH-L1, promoting a tumorigenic phenotype with an increase in proliferation, adhesion, cell migration and apoptosis inhibition [102]. 

This evidence shows that both EBV and KSHV contribute to the primary effusion lymphomagenesis and co-infection can increase the likelihood of tumor formation by shaping the tumor microenvironment and providing proliferation and survival advantages.

## 4. Modulation of Innate Immune Responses by EBV and KSHV

Human γ-herpesviruses, unlike viruses that only achieve acute infections, are not cleared by human immune responses, and establish latent infections [6,103]. It is a fine-tuned balance between the host immune responses and the pathogen immune evasion mechanisms that allows this persistence of EBV and KSHV without causing disease. This equilibrium, in which KSHV and EBV modulate the observed immune responses, was established during co-evolution over time and can be recognized in both innate and adaptive immunity to these viruses. Focusing on innate immunity, four classes of pathogen recognition receptors (PRR) are reported to be implicated in the recognition of EBV and KSHV: Toll-like receptors (TLR), RIG-I-like receptors (RLR), NOD-like receptors (NLR) and intracellular DNA-sensors like cGAS [104,105,106]. Activation of these pathways primarily leads to NF-κB-mediated production of inflammatory cytokines, induction of type I interferons (IFNs) or inflammasome activation and can be mediated by infected cells such as B cells, plasmacytoid dendritic cells (pDCs) and endo- and epithelial cells themselves. Apart from infected cells, activated monocytes, macrophages and classical dendritic cells (cDCs) harboring those PRR can also induce such responses. Despite knowledge of the involved pathways, there is no primary immunodeficiency (PID) affecting type I IFN responses described to predispose for γ-herpesviruses, and there remains a lot of open questions on how the innate immune sensing of both viruses influences the course of infection [107,108,109,110,111]. Along this line, it was shown that in vivo depletion of pDCs, even though being the main source of IFN after EBV infection, had only transient effects on EBV infection or on CD8^+^ T cell responses, which were thought to be primed by DCs [107]. Furthermore, pDCs are transiently depleted during symptomatic primary EBV infection in humans [112,113]. This insensitivity to type I IFN responses might be caused by the plethora of gene products of all γ-herpesviruses counteracting the above-mentioned immune responses reviewed in detail by Lange et al., stressing the importance to overcome early defense mechanisms for persistent infections [105]. In general, similar strategies are applied by both EBV and KSHV (Figure 2), all leading to the inhibition of PRR-mediated responses. In the first place, viral gene products may interfere with the expression of host proteins involved in PRR signaling cascades, either directly via viral miRNAs, by possessing exonuclease activity or by interacting with promoter sites to inhibit anti-viral gene expression [114,115,116,117,118,119]. So, for example, KSHV miRNA miR-K9 and miR-K5 can directly target MyD88, leading to reduced pro-inflammatory cytokine production and both KSHV LANA and kb-ZIP can abrogate IFN-*β* promoter activity [115,116,120]. Next, cellular proteins can be suppressed by expression of viral homologues, such as KSHV vIRF1-4 inhibiting the cellular interferon regulatory factors or KSHV ORF63 inhibiting inflammasome activation by NLRP1 mimicry [89,121,122,123]. In addition, viral phosphatases and kinases such as EBV BGLF4 can directly modulate enzyme activities thereby decreasing PRR downstream signaling [124,125,126]. In addition, viral proteins can modulate ubiquitylation and proteasomal degradation, exemplified by KSHV RTA, which possesses E3-ubiquitin ligase activity and targets, for example, MyD88 [127,128]. Finally, the direct interaction of gene products for both virus and host can prevent conformational changes or nuclear translocation, as it is the case for KSHV ORF45, which blocks the nuclear translocation of IRF7 [129,130]. Besides expressing viral immune evasions, the inexistence of protein expression of the EBV latency program 0 and the low expression level of all latent EBV proteins can be regarded as a hiding mechanism from human immune responses [104]. Overall, it still remains unclear if the viral pattern recognition in infected cells or bystander antigens present or viral sensing dendritic cells restrict EBV and KSHV infection. 

An additional line of early defense is mediated by innate immune cells such as NK, NKT and γδ T cells, whose phenotype might be directly shaped by the viral infection. Underlining the importance of NK cell responses in EBV infection are PIDs affecting NK cell differentiation, activating NK cell receptors or NK cell effector functions, but also the observed expansion of NK cells during infectious mononucleosis (IM) with regards to numbers and frequency [131,132,133,134]. Expanding NK cells are in an early differentiation state; CD56^dim^CD16^+/−^NKG2A^+^NKG2C^−^ and their protective function might be mediated either via activating NK cell receptors NKG2D and DNAM-1, via CD16-mediated antibody-dependent cellular cytotoxicity targeting lytically-replicating EBV, or via preventing B cell infection by direct removal of viral particles bound to the B cell surface [132,135,136,137,138]. Further differentiation driven by co-infection, in case of CMV into NKG2C^+^KIR^+^ adaptive NK cells, was shown to go along with impaired EBV-specific immune control [139]. Similarly, co-infection with KSHV is associated with further NK cell differentiation into CD56^−^CD16^+^CD39^+^ NK cells in humanized mice, an even less cytotoxic phenotype that might suppress immune responses via CD39 [31]. This reduced NK cell cytotoxicity is also observed in KS patients, which correlates with downregulated-activating NK cell receptors such as NKG2D, NKp30 or CD161 and with upregulation of the inhibitory receptor PD-1 [140,141,142]. Furthermore, KSHV gene products directly protect the infected cells by downregulating activating NK cell receptor ligands on their surface such as NKG2D ligands MICA/B, AICL, CD155 or Nectin-2 but also via secreting the viral chemokine vMIP-II blocking NK cell receptors involved in NK cell migration, such as CX3CR1 and CCR5 [143,144,145,146,147]. Therefore, early differentiated NK cells restrict lytic EBV infection, but KSHV co-infection compromises the cytotoxic function of these innate lymphocytes.

## 5. Adaptive Immune Responses to EBV and KSHV

Imbalance between host and pathogen can also be caused by deviations in the adaptive immune response and may lead to diseases such as IM in cases of overactive immune responses or to the development of malignancies or chronic active EBV in cases of lacking immune responses. Characteristics of patients affected by γ-herpesvirus associated malignancies include impaired cytotoxic responses, especially T cell responses [148]. Reasons may be primary immunodeficiencies (PID) affecting TCR signaling, costimulatory molecules and IFNγ signaling, but also co-infection with HIV, iatrogenic immunosuppression or advanced age [22,131].

The main cytotoxic effectors, the CD8^+^ T cells, highly expand in numbers during IM, the acute symptomatic primary EBV infection [133,149]. In IM, single EBV specificities can make up to 50% of the total CD8^+^ T cells during IM [150,151,152]. EBV specific T cells are primarily directed against immediate early (IE) gene products, to a lesser degree against early (E) gene products and even fewer against late (L) gene products, while latent antigen-specific T cells only make up around 0.1–0.5% and are mainly directed against the EBNA3 family of proteins [152,153,154,155]. The hierarchy of recognized antigens also remains during latent infection, although upon contraction of T cell numbers, the frequencies of EBV-specific CD8^+^ T cells decrease to 2% recognizing lytic and to 1% recognizing latent gene products, respectively [155,156,157,158]. The expanded CD8^+^ T cells during IM are of an activated phenotype being HLA-DR^+^CD38^+^CD69^+^Ki-67^+^ but lacking lymphoid homing markers such as CCR7 or CD62L, thus potentially explaining the low recruitment into tonsils resulting in lower EBV-specific T cell responses at the site of infection [150,154,159]. CD4^+^ T cells do not expand in numbers, yet EBV-specific responses increase to up to 1% of total CD4^+^ T cells and thereby contribute to increased overall activation of CD4^+^ T cells [133,160]. Contrary to CD8^+^ T cells, they are more often directed against latent antigens and may emerge delayed with EBNA1-directed responses arising only several months after primary infection [160]. EBV-directed CD4^+^ T cells can be cytotoxic and are mostly of a Th1-like phenotype expressing T-bet, IFN-γ, TNFα, Perforin and Granzyme B [161,162,163,164]. During asymptomatic primary infection, similarly high viral load levels as in IM patients were detected in a cohort of Gambian children, though without the accompanying CD8^+^ T cell expansion, questioning the protective effect of these cells during early years of life when seroconversion often occurs [165]. Nevertheless, successful adoptive transfer experiments of EBV-specific CD8^+^ T cells in lymphoma and PTLD patients, and depletion experiments in humanized mice leading to increased lymphomagenesis underline the protective value of EBV-specific CD8^+^ T cells [166,167]. Those lines of evidence are absent for KSHV-specific immunity. Epidemiology and PID predisposing for KSHV-associated diseases speak strikingly for an involvement of T cells, but experimental data are scarce [22,131]. In the early 21st century, substantial effort was put into identifying targeted epitopes, but only recent studies by Roshan and Nalwoga systematically investigated KSHV-directed IFN-γ responses on a proteome-wide scale [168,169,170]. Both studies showed very weak KSHV-directed T cell responses around 1 log lower in magnitude compared to EBV and CMV controls, and high heterogeneity between patients with no immunodominant antigen being recognized by most individuals. In addition, the amount of recognized KSHV antigen derived peptide pools of 1–5 per individuum differs greatly from EBV infection with a mean of 21 different proteins recognized per patient [155]. Earlier reported work on the predominant recognition of early and late lytic KSHV-antigens was based on trends seen in seven individuals only and does not seem to be confirmed in the study by Nalwoga et al. [168,171]. The hierarchy observed in responses towards EBV antigens might have to do with the direct priming of T cells by infected B cells and with evasions expressed in late lytic stages which simultaneously reduces the presentation of those genes, making it more unlikely to be recognized by T cells [155,172]. This leaves room for speculations that the lack of hierarchy in KSHV-directed responses might be a hint towards cross-primed responses that could be initiated by dendritic cells. 

In contrast to EBV, there is no severe or prototypic illness associated with primary KSHV infection allowing for the characterization of protective immune correlates [170,173,174]. Even though there are cases described in which mononucleosis or lymphadenopathy were associated with acute or reactivated KSHV infection, most reported patients suffer only from mild symptoms such as rashes or fever which are, contrary to EBV-related IM, not accompanied by a massive cytotoxic T cell expansion [173,174,175,176]. One of the first prospective studies characterizing the immune composition upon KSHV seroconversion showed no changes in T cell numbers or in phenotype, but occurrence of KSHV-directed IFN-γ responses along with KSHV viremia [173]. T cell responses seemed to peak only 1–2 years after seroconversion. Focusing more on chronic KSHV infection, another study observed no changes in *αβ* T cell subset frequencies, but a higher frequency of *γδ* V*δ*1 T cells in KSHV^+^HIV^−^ individuals compared to age-matched KSHV^−^ controls [177]. These *γδ* V*δ* T cells were strongly reactive against KSHV-infected PEL cell lines, which contrasts to what was observed for *αβ* T cells: in vitro experiments using CTL clones or Jurkat cells showed that PEL cell lines elicit only weak T cell responses, probably due to the interference of KSHV with MHC class I and II restricted antigen presentation [178,179,180]. However, implications of the impaired immunogenicity of PEL cell lines in a clinical setting remain unclear since most studies focusing on KSHV-directed T cell responses do not specifically focus on PEL patients but only differentiate between healthy and diseased virus carriers, including MCD and KS patients. Even there, data are somewhat contradictory with Roshan et al. reporting greater diversity in recognized antigens in diseased patients and earlier studies from Guihot and Lambert reporting the opposite with a greater diversity in healthy patients [169,181,182]. Nevertheless, all three studies demonstrated that in vitro, KSHV-restricted CD4^+^ and CD8^+^ T cells derived from healthy volunteers and diseased patients can be both mono- or polyfunctional, expressing IFN-γ, IL-2, CD107, MIP-1B and TNFα [169,183,184,185]. This cytokine profile is in agreement with PIDs affecting IFN-γ receptor or STAT4 that predispose for KS, with a KS tumor microenvironment in which PBMCs secrete high levels of Th1 cytokines, and also with KSHV encoded viral homologues of cellular chemokines, such as vMIPI-III, which counteract Th1 responses by rather skewing into a more Th2-like microenvironment for immune evasion [146,186,187,188,189,190]. On the other hand, the Th2-cytokine IL-5 was reported to be associated with better outcomes in KS, and pulmonary KS was shown to be associated with reduced expression of IFN-γ and other polyfunctional effectors mentioned before, therefore resulting in a reduced proinflammatory environment [191,192]. These discrepancies illustrate that the actual immune correlates conferring protection from KSHV-associated malignancies are still not identified, and while a number of studies have focused on CD8^+^ T cell-mediated IFN-γ responses, there are only a few studies investigating the importance of CD4^+^ or γδ T cell responses in KSHV infection.

## 6. Conclusions and Outlook

These immune responses against EBV and KSHV ensure co-existence without pathology in most persistently infected individuals. Therefore, it should be possible to re-establish immune control by vaccination in patients who suffer from EBV- and KSHV-associated pathologies or are at risk for these. The global disease burden of EBV- and KSHV-associated diseases, with yearly tumor incidences of 300,000 and 100,000, respectively, indeed argues for the development of EBV- and KSHV-specific vaccines [7,193,194]. Many of the respective vaccine efforts focus on the induction of neutralizing antibodies against EBV and KSHV [195,196,197,198,199,200]; even so, natural immunity is thought to be primarily mediated by cytotoxic lymphocytes [131,201]. Unfortunately, the recombinant viral vector vaccines to induce cytotoxic CD8^+^ T cell responses against EBV seem to be falling behind the neutralizing antibody-inducing vaccine efforts [202,203,204,205]. Nevertheless, an EBV-targeting vaccine will probably come into existence in the next few years and we will see how this can influence global disease burden by this human tumor virus.

Previously, it was shown that induction of neutralizing antibodies against EBV gp350, the vial envelope protein that mediates attachment via complement receptors (e.g., CD21) to human B cells, reduced the incidence of symptomatic primary EBV infection (infectious mononucleosis) by 78% [198,200]. Therefore, adolescents still seronegative for EBV and with a high risk to develop IM upon EBV infection [206,207] could benefit from a neutralizing antibody-inducing vaccine against EBV, if primary infection is thereby rendered asymptomatic and not only delayed. An increased risk for EBV-associated Hodgkin’s lymphoma and the autoimmune disease, multiple sclerosis (MS), has been observed after IM [208,209,210]. Multiple sclerosis affects more than 2 million individuals worldwide [211]. Therefore, vaccine-induced EBV neutralizing antibodies could reduce these risks for EBV-associated diseases at the same time as IM. However, in comparison to the 32-fold increased risk for multiple sclerosis by EBV infection in general [212], the 2-fold increased risk after IM compared to asymptomatic primary infection is rather modest. Nevertheless, a better understanding of the mechanistic contribution of EBV infection to MS development would enable us to assess if EBV-specific vaccination could influence this autoimmune disease. As EBV also seems to contribute to KSHV persistence and KSHV-associated tumor burden in the case of PEL, vaccination against EBV might also prove beneficial with regards to KSHV infection. KSHV-specific vaccination efforts might also significantly reduce KSHV-associated disease burden [7]. Low prevalence of this tumor virus in Middle and Northern Europe as well as North America might suggest that establishing robust immunity against KSHV by vaccination could achieve low prevalence of KSHV in Sub-Saharan Africa and Southern Europe. One would predict that this would significantly reduce the disease burden by KSHV. Therefore, robust immune control in most EBV and KSHV carriers suggests that vaccines should be developed that reinstate this immune control in patients who suffer from diseases that are associated by these two oncogenic human γ-herpesviruses.

## Figures and Tables

**Figure 1 viruses-14-02709-f001:**
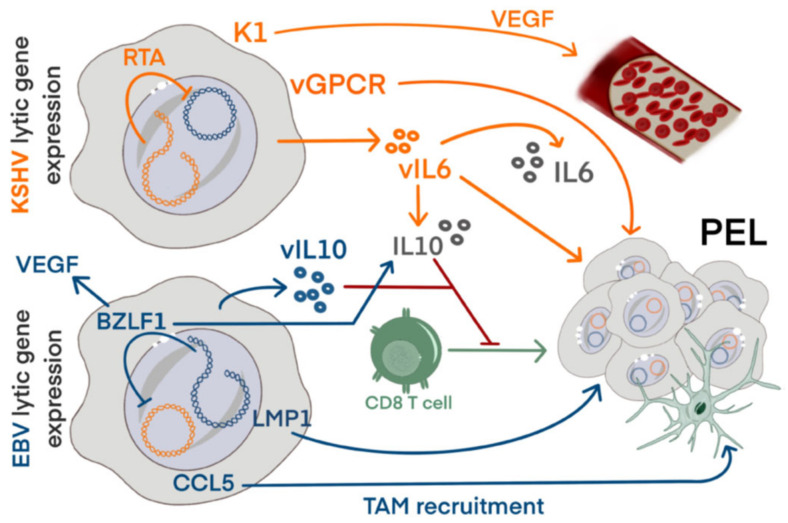
Expression of lytic EBV and KSHV genes can condition the tumor microenvironment. Primary effusion lymphoma (PEL) is associated with KSHV, however 90% of tumors also carry EBV. EBV and KSHV most likely contribute to the tumor environment simultaneously through their lytic gene expression. Lytic KSHV expression contributes through expression of K1, which promotes expression of VEGF and angiogenesis. viral G-protein coupled receptor expression promotes proliferation. Expression of the viral cytokine vIL6 promotes production of IL6 and IL10 and increases PEL proliferation. EBV lytic gene expression contributes through CCL5 production that attracts monocytes, which as tumor associated macrophages (TAM) have immune suppressive functions. Expression of viral IL10 can suppress CD8^+^ T cell responses.

**Figure 2 viruses-14-02709-f002:**
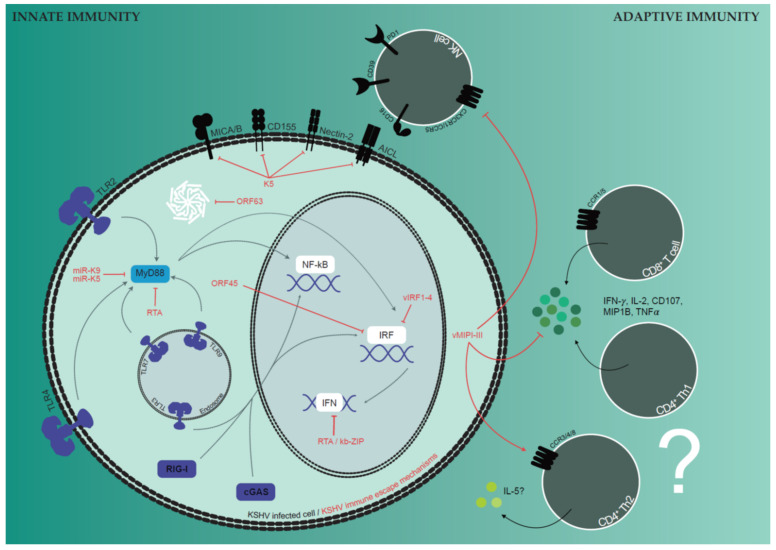
Balance between host immune responses and viral immune modulation mechanisms allow persistence of KSHV. TLR, RLR, NLR and intracellular DNA-sensor cGAS are the four PRRs reported to sense KSHV infection (blue) and to induce NF-κB-mediated inflammatory cytokine production, type I IFN response and inflammasome activation (white). KSHV immune evasions (red) counteract PRR-induced signaling pathways via different means, e.g., via reducing the expression of signaling proteins (miR-K9/K5, RTA, kb-ZIP), via suppression of cellular proteins by viral homologues (vIRF1-4, ORF63), via targeting signaling proteins for proteasomal degradation (RTA) or via inhibition of nuclear translocation of signaling proteins (ORF45). Cellular innate immune response is modulated by reducing cytotoxicity of NK cells via driving differentiation into a late phenotype characterized by CD39 expression and loss of NKG2D, via downregulation of activating NK cell receptor ligands and via inhibiting NK cell migration by viral chemokine secretion. IFN-γ derived from NK cells, CD8^+^ or Th1 CD4^+^ T cells might protect from KSHV-associated malignancies, although T cell correlates conferring protection from KSHV-associated malignancies are not fully understood.

## Data Availability

Not applicable.

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
