# Peer review of "Co-Infection of the Epstein–Barr Virus and the Kaposi Sarcoma-Associated Herpesvirus"

_viruses, 2022, doi:10.3390/v14122709_

Round 1

Reviewer 1 Report

This is a brilliant review regarding the concurrent role of two gamma herpesviruses associated with lymphoid malignancies, namely primary effusion lymphoma (PEL). The discussion how EBV and KSHV altogether are involved in persistence and lymphomagenesis is sustained by a corpus of solid references.

The authors could put more emphasis on the primordial role of the BZLF1 gene of EBV  encoding Zta or ZEBRA protein, in the process of lymphomagenesis associated with EBV, particularly in figure 1 and in the text. Indeed this viral transcription factor is able to induce the secretion of VEGF, IL-10  (Jones RJ et al. Int J Cancer. 2007 Sep 15;121(6):1274-81. doi: 10.1002/ijc.22839. PMID: 17520680- Hong GK, et al. J Virol. 2005 Nov;79(22):13984-92. doi: 10.1128/JVI.79.22.13984-13992.2005. PMID: 16254334; PMCID: PMC1280197 - Mahot S et al. J Gen Virol. 2003 Apr;84(Pt 4):965-974. doi: 10.1099/vir.0.18845-0. PMID: 12655098), and therefore can be an prominent factor of tumor progression in this context (Habib M, et al.Sci Rep. 2017 Sep 5;7(1):10479. doi: 10.1038/s41598-017-09798-7. PMID: 28874674; PMCID: PMC5585268 - Germini D, et al. Cancers (Basel). 2020 Jun 5;12(6):1479. doi: 10.3390/cancers12061479. PMID: 32517128; PMCID: PMC7352903.

Author Response

We thank both reviewers for their constructive comments which we have implemented into the revised manuscript version. These are marked as tracked changes in the revised manuscript and outlined below.

Reviewer #1

This is a brilliant review regarding the concurrent role of two gamma herpesviruses associated with lymphoid malignancies, namely primary effusion lymphoma (PEL). The discussion how EBV and KSHV altogether are involved in persistence and lymphomagenesis is sustained by a corpus of solid references.

The authors could put more emphasis on the primordial role of the BZLF1 gene of EBV encoding Zta or ZEBRA protein, in the process of lymphomagenesis associated with EBV, particularly in figure 1 and in the text. Indeed this viral transcription factor is able to induce the secretion of VEGF, IL-10  (Jones RJ et al. Int J Cancer. 2007 Sep 15;121(6):1274-81. doi: 10.1002/ijc.22839. PMID: 17520680- Hong GK, et al. J Virol. 2005 Nov;79(22):13984-92. doi: 10.1128/JVI.79.22.13984-13992.2005. PMID: 16254334; PMCID: PMC1280197 - Mahot S et al. J Gen Virol. 2003 Apr;84(Pt 4):965-974. doi: 10.1099/vir.0.18845-0. PMID: 12655098), and therefore can be an prominent factor of tumor progression in this context (Habib M, et al.Sci Rep. 2017 Sep 5;7(1):10479. doi: 10.1038/s41598-017-09798-7. PMID: 28874674; PMCID: PMC5585268 - Germini D, et al. Cancers (Basel). 2020 Jun 5;12(6):1479. doi: 10.3390/cancers12061479. PMID: 32517128; PMCID: PMC7352903.

We thank the reviewer for his/her comments and have added the suggested information to our revised manuscript (from line 142 on).

Reviewer 2 Report

In this manuscript, the authors reviewed the mechanism of KSHV and EBV co-infection in B cells, as well as the innate and adaptive immune response after KSHV and EBV co-infection. In addition, they point out that the presence of EBV contributes to the existence of a persistent latent form of KSHV in B cells. In primary effusion lymphoma, EBV and KSHV regulate the tumor microenvironment through the expression of cleavage genes. Most importantly, understanding the asymptomatic persistence of these two tumor viruses in vivo will help to develop vaccines to prevent and treat EBV and KSHV infection. This article summarized the immune response of EBV and KSHV co-infection, which is of great value in the diagnosis, treatment and prevention of related diseases.

It is suggested to add in the revised version:

1. Molecular mechanism of EBV reactivation;

2. The mechanism and immune response of secondary EBV infection after KSHV infection;

3. How to determine whether PEL is directly related to KSHV or EBV?

Minor:

1. In line 276 of adaptive immunity, the author refers to hierarchy. What does this hierarchy specifically refer to? Has it been shown during primary EBV infection?

2. In the part of adaptive immunity, the author described the IM immune response caused by primary EBV infection and KSHV infection respectively. Since the previous article and title described the co-infection of the two viruses, what is the relationship between IM and KSHV infection? Is there IM associated with KSHV infection?

3. As described in the manuscript, EBV and KSHV are co-infected in primary effusion lymphoma. What are the innate and adaptive immune responses in PEL patients?

4. Abbreviations in the manuscript are confusing. For example, in line 92 of the persistence of KSHV in EBV-infected B cells, the abbreviation appears for the first time without a full name.

Author Response

We thank both reviewers for their constructive comments which we have implemented into the revised manuscript version. These are marked as tracked changes in the revised manuscript and outlined below.

Reviewer #2

In this manuscript, the authors reviewed the mechanism of KSHV and EBV co-infection in B cells, as well as the innate and adaptive immune response after KSHV and EBV co-infection. In addition, they point out that the presence of EBV contributes to the existence of a persistent latent form of KSHV in B cells. In primary effusion lymphoma, EBV and KSHV regulate the tumor microenvironment through the expression of cleavage genes. Most importantly, understanding the asymptomatic persistence of these two tumor viruses in vivo will help to develop vaccines to prevent and treat EBV and KSHV infection. This article summarized the immune response of EBV and KSHV co-infection, which is of great value in the diagnosis, treatment and prevention of related diseases.

It is suggested to add in the revised version:

  1. Molecular mechanism of EBV reactivation;

We have now expanded on lytic EBV reactivation from latency 0/I upon plasma cell differentiation, citing studies that document BLIMP1 and XBP1 mediated BZLF1 induction (line 44-46).

  1. The mechanism and immune response of secondary EBV infection after KSHV infection;

We thank the reviewer for this comment. To our knowledge, there are no patient or in vivo studies that address this. There is a study by Fauré et al 2019 that has shown the time sensitivity for the co-infection, which showed that in vitro dual infection of B cells is most efficient when EBV co-infection occurs 24h post KSHV infection. If EBV co-infection occurs more than 24 h before or after KSHV infection, successful establishment of indefinetly growing, dual-infected B cell lines is reduced. The limited patient studies also do not allow to draw conclusions about secondary EBV infection after KSHV infection, mostly because most of the observed patient cohorts already show EBV infection at a young age.

  1. How to determine whether PEL is directly related to KSHV or EBV?

We thank the reviewer for this comment and have clarified this in the manuscript (from line 66 on).

Minor:

  1. In line 276 of adaptive immunity, the author refers to hierarchy. What does this hierarchy specifically refer to? Has it been shown during primary EBV infection?

EBV antigens were observed to be recognized in a certain hierarchy during infectious mononucleosis, the acute primary EBV infection. For clarification, we specified in line 291 that this hierarchy refers to the “recognized antigens”.

  1. In the part of adaptive immunity, the author described the IM immune response caused by primary EBV infection and KSHV infection respectively. Since the previous article and title described the co-infection of the two viruses, what is the relationship between IM and KSHV infection? Is there IM associated with KSHV infection?

We added a paragraph from line 329-334 in order to clarify, that IM is in most cases not related to KSHV and that the IM immune response, specifically the massive CD8+ T cell expansion, is not observed upon KSHV seroconversion.

  1. As described in the manuscript, EBV and KSHV are co-infected in primary effusion lymphoma. What are the innate and adaptive immune responses in PEL patients?

We thank the reviewer for this comment. Unfortunately, we are not aware of any primary research papers that are specific enough to answer this question. Epidemiological data shows, that PEL is primarily a disease of immunocompromised patients but clinical and immunological data of PEL patients is scarce, with most studies not distinguishing in diseased patients between MCD, KS or PEL patients. For clarification, we added a paragraph from line 338 to 354, including data on T cell recognition of PEL cell lines in vitro.

  1. Abbreviations in the manuscript are confusing. For example, in line 92 of the persistence of KSHV in EBV-infected B cells, the abbreviation appears for the first time without a full name.

We thank the reviewer for this comment and have adapted all abbreviations. As for the two latent promoters Qp and Cp, we are not aware of a full name of these promoters but have clarified their function in lines 103-104.

Round 2

Reviewer 2 Report

none